# Route of administration significantly affects particle deposition and cellular recruitment

Keziyah Yisrael[1,2☯], Ryan W. Drover[3,4☯], Malia L. Shapiro[1,2], Martha Anguiano[1], Nala Kachour[1,2], Qi Li[3,4], Emily Tran[1], David R. Cocker, III[3,4], David D. Lo[1,2,5]*

1 Division of Biomedical Sciences, University of California, Riverside School of Medicine, Riverside, California, United States of America, 2 BREATHE Center, University of California, Riverside, Riverside, California, United States of America, 3 Department of Chemical and Environmental Engineering, University of California, Riverside, Riverside, California, United States of America, 4 College of Engineering-Center for Environmental Research and Technology (CE-CERT), University of California, Riverside, Riverside, California, United States of America, 5 Center for Health Disparities Research, University of California, Riverside, Riverside, California, United States of America

☯ These authors contributed equally to this work.
* david.lo@medsch.ucr.edu

**Data Availability Statement:** All data and analysis are presented in the paper. No additional raw data are held in a public repository.

## Abstract

Lung exposures to dusts, pollutants, and other aerosol particulates are known to be associated with pulmonary diseases such as asthma and Chronic Obstructive Pulmonary Disease. These health impacts are attributed to the ability of aerosol components to induce pulmonary inflammation, which promotes tissue remodeling, including fibrosis, tissue degradation, and smooth muscle proliferation. Consequently, the distribution of these effects can have a significant impact on the physiologic function of the lung. In order to study the impact of distribution of inhaled particulates on lung pathogenesis, we compared the effect of different methods of particle delivery. By comparing intranasal versus aerosol delivery of fluorescent microspheres, we observed strikingly distinct patterns of particle deposition; intranasal delivery provided focused deposition concentrated on larger airways, while aerosol delivery showed uniform deposition throughout the lung parenchyma. Recognizing that the impacts of inflammatory cells are contingent upon their recruitment and behavior, we postulate that these variations in distribution patterns can result in significant alterations in biological responses. To elucidate the relevance of these findings in terms of biological representation, we subsequently conducted an investigation into the responses elicited by the administration of endotoxin (bacterial Lipopolysaccharide, or LPS) in a transgenic neutrophil reporter mouse model. As with the microsphere results, patterns of recruited neutrophil inflammatory responses matched the delivery method; that is, despite the active migratory behavior of neutrophils, inflammatory histopathology patterns were either focused on large airways (intranasal administration) or diffusely throughout the parenchyma (aerosol). These results demonstrate the importance of modes of aerosol delivery as different patterns of inflammation and tissue remodeling will have distinct impacts on lung physiology.

**Funding:** The research presented in this publication was supported by the National Institute On Minority Health And Health Disparities of the National Institutes of Health under Award Number U54MD013368 to DDL. The funders had no role in study design, data collection and analysis, decisions to publish, or preparation of the manuscript.

**Competing interests:** The authors have declared that no competing interests exist.

# Introduction

Despite its position as an internal organ, the lung provides a major interface with the environment, with a high volume of airflow across its large surface area. While it provides the critical surface for gas exchange, it also comprises a major mucosal barrier against aerosol particulates, toxins, and infectious organisms. The anatomy of the lung must provide an efficient path for airflow during respiration; yet also enable immune system mechanisms for maintaining clear airways and alveoli. Innate immune cells such as resident alveolar macrophages provide an important scavenging function, but the warm moist environment can promote invasion or colonization by infectious microbes that are not as easily eliminated [1]. Thus, lung tissue must enable the efficient recruitment of blood-borne inflammatory cells including neutrophils, monocytes, and lymphocytes, as well as mechanisms to ensure effective clearance of the products of inflammation [2,3]. Pulmonary diseases demonstrate the limits of these mechanisms, and studies on the pathogenesis of various pulmonary inflammatory diseases are dependent on disease models that replicate the physiological mechanisms protecting the lung.

In addition to lung resident immune cells, this organ also provides a partial physical barrier via the branched structure of the airways [4–6]. These bifurcations represent a partial anatomical obstruction impeding the flow of particles and microbes from making it into the distal airways. Although the threat of adverse reactions to inhaled particulate matter varies depending on their size and content, airborne particles can elicit some degree of an inflammatory response. Airborne particulate matter (PM) refers to a complex mixture of aerosols, which are present in the ambient air. PM is characterized by particle diameter with particles with a diameter of 10 microns or larger categorized as "coarse", particles $\leq$ 10 microns are referred to as $PM_{10}$, and those which are $\leq$ 2.5 microns are referred to as $PM_{2.5}$. Particles which fall into the $PM_{2.5}$ category are of most concern as these particles are able to reach deep into the distal lung parenchymal tissue and airways and induce adverse health effects [7]. Many studies have shown links to health effects in as little as 24 hours with exposure to $PM_{2.5}$ particulates [8]. A study done by Wang et. al revealed that exposure to $PM_{2.5}$ was associated with risks of cardiovascular disease, respiratory illness, as well as some forms of cancer [9]. These particles can be directly emitted from a source such as motor vehicle exhaust or from chemical reactions of gases within the atmosphere. Frequent particulate matter exposure has also been linked to various health conditions such as asthma exacerbations, decreasing lung function, increased irritation of airways, coughing, and difficulty breathing. Due to the frequency of exposure to particulate matter, many studies have investigated these exposures and the related health impacts.

Most in vivo studies regarding PM and aerosol exposures have been done through the traditional intranasal administration method. However, this episodic method of exposure is not representative of chronic human exposure to particulates. In addition, it is difficult to ensure that all liquid is inhaled through the nasal passages and deposited directly into the lung; this may not only lead to inaccurate estimates of delivered dose but may also have off target effects as much of the solution may be swallowed and deposited in the digestive tract. Moreover, intranasally administered particle suspensions are likely to get fixed in the upper respiratory tract and may not penetrate the airways, leading to inaccurately skewed results. To address these limitations, we developed an environmental exposure chamber to study health impacts of suspended PM exposure [10]. To verify the efficacy of our chamber, we studied the impacts of differences in particle deposition in response to the method of administration. The results of this study elucidate the significance of method of administration on the pattern of particle deposition and cellular recruitment.

## Materials and methods

### Microspheres

Ready-made 1-micron FluoSpheres Carboxylate-Modified microspheres (FluoSpheres, Molecular Probes) were used to assess particle deposition as a result of exposure method. These polystyrene microspheres are loaded with red fluorescent dye, detectable with fluorescence microscopy at a wavelength of 580-605nm. As these microspheres are biologically inert, we used these to characterize the simple distribution of particle deposition in the lung of mice without any superimposed biological effects.

### Aerosol exposure chamber

Aerosol exposures were conducted using dual animal chambers as developed and characterized in Peng et al. (2019) and used in Biddle et al. (2021, 2023) [10,11]. The conditions in the chambers were monitored for the duration of the exposures, including relative humidity, temperature, and atmospheric pressure. Aerosol-exposed mice were administered aerosols generated from an aqueous solution (of the chosen aerosol, microspheres or LPS, at a concentration determined to achieve the target aerosol concentration within the chamber), dried by two in-line silica gel columns (3.5–4.5 LPM) mixed with dry filtered air (0.5–1.0 LPM) to balance chamber volume exchange rates, as previously applied in Peng et al. Particulate matter in the chamber was monitored using a scanning mobility particle sizer (SMPS, including Series 3080 Electrostatic Classifier and Ultrafine Condensation Particle Counter 3776, TSI) and a laser aerosol spectrometer (LAP 323, Topas GmbH) to assist in maintaining stable and repeatable environmental PM exposure conditions.

As discussed in Peng et al. (2019), the animal exposure chambers provide the ability to expose mice to a targeted aerosol for a determined duration of time, while they can move and live unimpeded by the experimental procedures. The chambers are characterized to ensure a well-dispersed distribution of aerosol throughout the chamber so that a given volume of air inhaled by a mouse is equivalent in aerosol concentration to any other pocket of air inhaled by other mice, throughout the duration of an exposure period and across multiple exposure tests. Beyond controlling the aerosol concentration, deposition in the lungs of the mice is subject to additional factors including obstruction by whiskers, nasal passages, bronchial passage bifurcation, and airway anatomy, but are consistent across exposure methods. In this study 1-micron microspheres were used in order to minimize the variance of these effects for evaluating deposition, as physical properties including the size, shape, hygroscopicity, volatility, and the electrostatic charge of the particle are maintained, while particle coagulation can be measured by the PM instruments to confirm that particulate size does not significantly vary during the exposure. The size of the 1-um particles also is in the size range to remain suspended in the air column, minimally affected by the anatomy of the airways and able to reach the alveolae largely unimpeded. This provides for consistency in the repeated exposures in this study; although the exposure (environmental aerosol presence through the exposure duration) is held constant, accumulated deposition in the lungs of the mice (delivered dose) cannot be directly measurable in real-time.

### Mouse models and exposure conditions

All animal procedures were performed according to protocol AUP #20210011 approved by the UCR IACUC and consistent with institutional and NIH guidelines. 8-week-old C57BL/6J (B6) mice were obtained from The Jackson Laboratory. Mice were exposed to the microspheres by either method of administration at a concentration of 1.71E+09/mL for the duration of

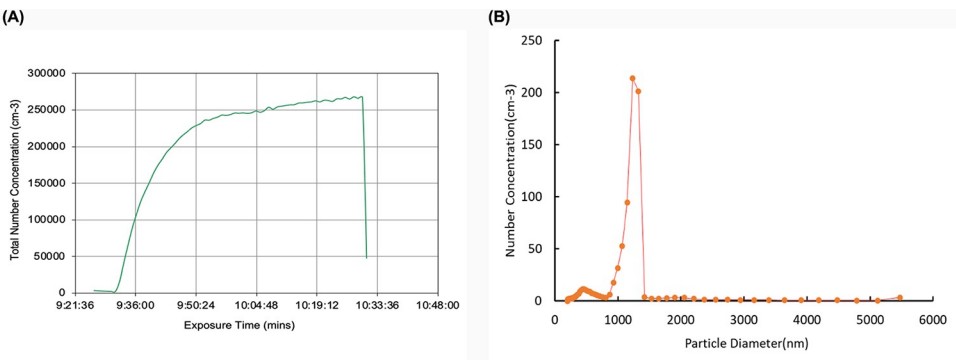

**Fig 1. Data collected during exposure duration to fluorescent microspheres. (A)** Total number concentration of particles within 0.5–1.3μm. **(B)** Average size distribution of particles injected into chamber.

1-hour. For environmental chamber aerosol exposures, a suspension of microspheres was continuously injected through an atomizer nozzle, and then passed through two drying columns on their way into the environmental chamber. Continuous monitoring of the aerosol suspension in the chamber (Fig 1A) confirmed that the microspheres were in a single-particle suspension with a measured monodisperse particle size distribution of 1-micron (Fig 1B).

Mice exposed via the aerosol chamber were continuously exposed for 1-hour and permitted to rest for 1-hour before processing. Intranasally exposed mice were given one dose of 40 μl of the solution containing the microspheres and were allowed to rest for 1-hour prior to processing. To alleviate suffering, animals were placed in an isoflurane chamber before cervical dislocation and processing. Mice were also intranasally exposed to one dose of 20 μl, alternating nostrils and administering 5 μl at a time. Another group of intranasally exposed mice were given one large dose of 20 μl at a single time. All mice were permitted to rest for 1-hour before sacrifice. Upon sacrifice, left lobes were collected and analyzed via fluorescence microscopy.

PGRP-S-dsRed transgenic reporter mice rely on the peptidoglycan recognition protein-S (PGRP-S) promoter sequences driving expression of the dsRed express2 coding sequence [12]. In this mouse model, cells such as neutrophils, eosinophils, and epithelial M cells will fluoresce red upon detection due to their regulated expression of PGRP-S. These mice were exposed to LPS via either exposure method and differences in distribution of dsRed expressing cellular recruits were assessed via Python software. Mice exposed to LPS intranasally were administered a concentration of 75 μg/mL. Mice were administered 40 μl doses at two time points 6 hours apart and were allowed to rest for 24 hours before sacrifice. This method was established through optimization of both the time and concentration for intranasal administration. Mice exposed via the aerosol chamber were exposed to a concentration of 15 μg/m$^3$ for the duration of 24 hours and sacrificed after 1-hour of rest. Mice were sacrificed in the same manner as described for the microsphere exposures.

Mice utilized for all aerosol exposures, were housed 2 per cage in a home cage lined with a thin layer bedding. Mice were supplied ample food and water and were allowed to freely drink and eat in their home cages. The mice were housed within their home cages inside of the larger exposure chamber used for aerosol delivery, equipped with an automatic day/night light cycle. Intranasally exposed mice were housed identical to the chamber cohort.

## Histology

For hematoxylin and eosin staining (H&E), mice were exposed to lipopolysaccharide under the aforementioned conditions; however, for this processing, these mice were sacrificed before

performing intratracheal instillation to inflate lungs with 0.3mL of a 1:1 mixture of optimal cutting temperature compound (OCT) and phosphate-buffered saline (PBS). These lungs were then dissected and flash frozen. Cryostat sectioning was performed on the fresh frozen tissue followed by H&E staining to assess cellular infiltrations.

Neutrophil recruitment was confirmed through the utilization of spinning disc confocal microscopy in conjunction with the Keyence system, as described below. The procedures employed for this approach were identical to those described previously for the microsphere exposure cohort. However, the specimens under examination only received DAPI nuclear staining to visualize the multilobed nuclei of the recruited neutrophils. As previously mentioned, based on the transgenic reporter in these mice, dsRed+ cells were detected and recorded within the Texas Red channel.

## Microscopy

A Keyence BZX800 and a custom Yokogawa/Zeiss spinning disc confocal microscope setup were used to image both microsphere deposition as well as dsRed positive cells in either study. All images were taken using either 10x or 20x objectives and tiled together to generate the whole lobe images; scale bars are included in the images. The larger stitched images were then used for analysis using a custom Python script.

## Analysis

Automated image analysis was used to quantify fluorescent microsphere and cell distribution throughout the tissue analysis. A stand-alone script was built using the IPython kernel (Python (3.9.7)) in JupyterLab (3.2.1, Jupyter) through Anaconda Navigator (2.1.4, Anaconda Inc.), with the libraries Matplotlib (3.6.0), NumPy (1.11.3), pandas (1.5.1), Python Imaging Library (Pillow/PIL, 9.3.0), and OpenCV (4.6.0.66) used to support functionality without modification to the image files from the Keyence microscope (.TIFF file formatting) [13]. This script was paired with the open-source image processing platform ImageJ (NIH) in order to preliminarily identify neutrophils during the PGRP experimental portion of the study [14]. Images were minimally processed at all stages of analysis to minimize manipulation biasing the analysis.

Preliminary analysis performed on fluorescent microsphere images centered on color analysis, identifying, and quantifying characteristic colors present to differentiate non-fluorescent tissue, auto-fluorescent tissue (as commonly present near airways), and fluorescent microspheres. These separations provided exclusionary criteria for identifying microspheres in the final analysis of measuring localized deposition to evaluate the dispersion of deposition within the lung tissue. The whole-lung analysis was performed by 1) creating a *mask* image to filter the characteristic color of fluorescent microspheres, 2) overlaying a grid of square cells a given size (100, 200, or 500 um), which was used 3) to subdivide the image into a numerical output based on the measured intensity of the fluorescent microspheres (as filtered for in (1)). 4) Following this, the tissue was re-analyzed with the *mask* image in (1) modified to include all color characteristic of tissue coverage. 5) These numerical outputs were aligned, and the output values from (4) were used to determine the proportion of tissue coverage within given grid cells. 6) Finally, the tissue coverage of a grid cell from (4) and (5) was used to include or exclude a given grid cell from analysis to ensure that tissue being analyzed was sufficiently representative of lung tissue, rather than biasing the analysis due to the inclusion of non-tissue space (such as airways or external non-lung space) that would not contain fluorescent microspheres.

Grid cell sizing and use for analysis was evaluated using combinations of three grid sizes (100 um, 200 um, 500 um square grids) and four tissue-coverage (50%, 75%, 90%, 98% tissue coverage within a given grid cell) criteria for the inclusion in the analysis of each lung image.

The grid sizing of 200 um with the tissue coverage criteria of 90% was used in the final image analysis as these criteria worked best to avoid biasing measurement through the inclusion of airway and non-tissue image areas. Although all size ranges tested yielded similar results, this selection criteria provided the most robust and repeatable analysis across the dataset.

Additionally, to identify distribution of particles and cells proximal to large and medium airways, deposition of microspheres and neutrophils within 200 um of the airways was evaluated as a fraction of the total deposition/presence in the lung.

## Interpretation of data

The index of dispersion (D, the ratio of the variance to the mean $D = \sigma^2/\mu$) was used as the primary statistic for comparison of the deposition throughout the lung tissue in this dataset [15]. The index of dispersion is a useful measure of comparison between two datasets with large differences in means to account for the larger variance due to larger means. In interpreting the index of dispersion, the threshold of $D = 1$ is commonly used, as Poissonian distribution has an equal variance and mean, resulting in an index of dispersion of $D = 1$. In the case of the random distribution of particles (or diffusion; Brownian motion; deposition in this study) the distribution of particles in a given volume is Poissonian. The index of dispersion can be used to assess a given pattern of distribution by dividing the space into equally sized segments, within each of which particles are counted, and the index of dispersion across the total space is calculated. An index of dispersion $D \gg 1$ commonly denote a clustered distribution, while an index of dispersion $D < 1$ indicates low variance (particularly with regard to outliers/extremes that notably impact the measured variance) throughout the space.

The coefficient of variation ($c_v$, the ratio of the standard deviation to the mean $c_v = \sigma/\mu$, or in a sample, the ratio of the sample standard deviation to the sample mean $c_v = s/x$) was used as an initial statistic for corroborating the measure of dispersion alongside the index of dispersion. While the standard deviation is commonly reported in evaluating the dispersion of numerical data, the variance (as used in the index of dispersion) weights outliers more heavily than data very near to the mean (as in the standard deviation). Due to the need for measuring extreme areas of deposition in the dataset, the standard deviation (and so $c_v$) obscured the extremes of the deposition in comparison to the index of dispersion enabling the emphasis in identification and measurement of this pattern of deposition.

## Results

To model typical aerosol particulate exposures, we exposed B6 mice to 1μm fluorescent microspheres via either intranasal administration or using the aerosol exposure chamber. This chamber is unique as it allows for continuous exposure to a given concentration of aerosols for any desired length of time. The chamber allows for uninterrupted aerosol exposure in the form of natural, unassisted ventilation. The chamber also allows for continuous monitoring of particle size distribution and PM mass concentration. With this chamber system, we mimic natural inhalation of particles as a physiologically natural system to study aerosol exposure. We hypothesized that exposure via the chamber will generate uniform distribution of particles throughout the lung while intranasally exposed mice will demonstrate nonuniform microsphere deposition. Results shown in Table 1 as well as Fig 2 illustrate the differences in the overall pattern of particle distribution between exposure methods.

### Method of administration alters microsphere deposition

Mice exposed to microspheres via the exposure chamber (Fig 3A) revealed an average index of dispersion of 0.627 as compared to animals subjected to exposure via intranasal instillation

**Table 1. Summary of results.**

| Exposure Condition | Method of administration | Index of dispersion | Average Variance | % Peri-airway Deposition |
|---|---|---|---|---|
| Microsphere exposure | Aerosol chamber | 0.627 | 0.266 | 26.48% |
| Microsphere exposure | Intranasal administration | 41.90 | 2043.0 | 52.68% |
| Lipopolysaccharide exposure | Aerosol chamber | 0.981 | 1.224 | 65.68% |
| Lipopolysaccharide exposure | Intranasal administration | 0.509 | 2.833 | 75.60% |

(Fig 3B), which had an average index of dispersion of 41.90 (Fig 3D). There is a significant increase in index of dispersion in mice administered the microspheres via intranasal administration when compared to the chamber group. The index of dispersion illustrated in Fig 3D indicates that the dispersion of particles due to administration via exposure chamber follows a uniform distribution pattern. When compared to the index of dispersion for mice exposed via the I.N. method, intranasally exposed mice showed a distinct, nonuniform, clustered phenotype. In addition to significant differences in the index of dispersion, overall variance between exposure groups also varied significantly (Fig 3E). Analysis revealed that chamber exposed mice demonstrated an average variance of 0.266 while I.N. exposed mice exhibited an average variance of 2043.0.

The calculated index of dispersion values revealed that mice exposed to the 1μm fluorescent spheres in the exposure chamber showed uniform distribution with particle deposition occurring in an even pattern throughout lung parenchyma. By contrast, animals exposed via intranasal instillation, demonstrated a pattern of sporadic, nonuniform deposition (Figs 3 and 4). Not surprisingly, it was also noted that specific patterns of particle distribution varied greatly from subject to subject in intranasally exposed animals.

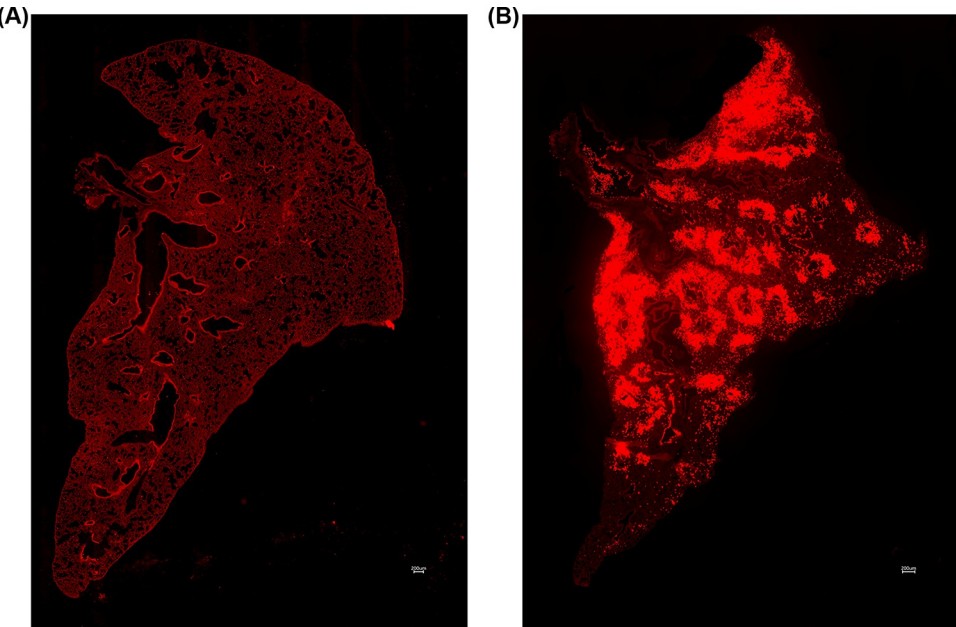

**Fig 2.** **(A)** 10x tile stitched fluorescence image of WT mouse exposed to 1μm fluorescent microspheres via exposure chamber. **(B)** 10x tile stitched fluorescence image of WT mouse exposed to 1μm fluorescent microspheres via intranasal administration.

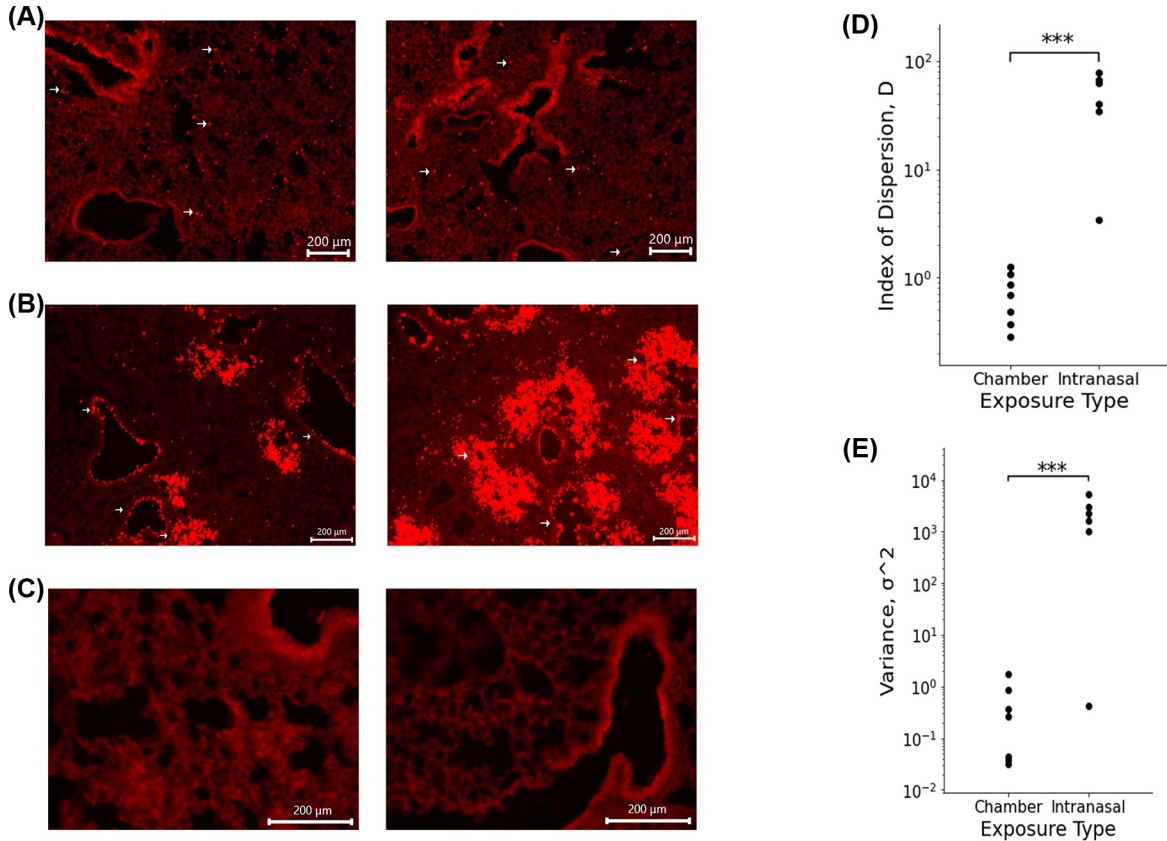

**Fig 3.** **(A)** Fluorescence images of microsphere deposition of WT mouse exposed to 1μm fluorescent microspheres via exposure chamber. Arrows highlight microspheres. **(B)** Fluorescence images of microsphere deposition of WT mouse exposed to 1μm fluorescent microspheres intranasally. Arrows highlight preferential clustering of microspheres around medium sized airways. **(C)** Fluorescence images of control WT mouse lung. **(D)** Index of Dispersion of chamber and intranasally exposed WT mice. **P value = 0.036581 I** Raw variance of chamber and intranasally exposed WT mice. **p value = 0.01363**.

Histologically, we noted significant differences in the general pattern of deposition of the microspheres between exposure groups. In mice exposed to the microspheres via intranasal administration, we noted significant microsphere deposition concentrated around medium and large airways; a feature exclusive to this administration method (Figs 2B, 3B and 4). I.N. exposed mice also frequently demonstrated deposition of microsphere mini-aggregates at terminal portions of the airways where they empty into the smaller alveolar spaces of the lung. In these mice, the microspheres distribution was patchy, and particles were inconsistently distributed throughout the lung when compared to chamber exposed animals (Fig 3). Chamber exposed mice demonstrated uniform deposition in which no clumping was observed neither around airways nor throughout the parenchyma itself (Fig 5). These animals demonstrated uniform delivery of the particles with broader coverage throughout the parenchyma, where nearly every region of the lung tissue contained deposited microspheres. In contrast, I.N. exposed animals showed significant gaps in deposition where large portions of the lungs had very minimal to no microsphere deposition. Another distinction in intranasally exposed mice is that there was frequent deposition within the lumen of airways; this feature was not observed in chamber exposed animals. This may be due to the preferential clustering of microspheres in and near the airway in liquid drops with aggregated microspheres, or simply due to the concentrated delivery of the particles, forcing some microspheres to be trapped within the airways.

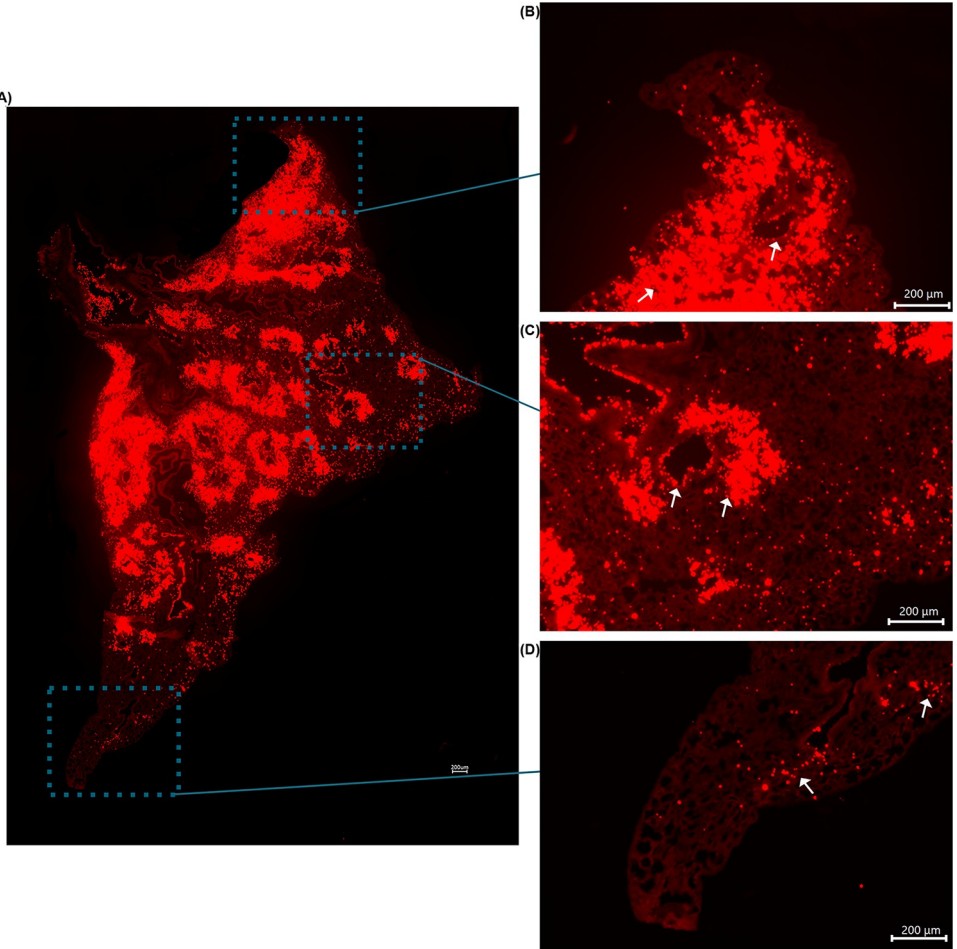

**Fig 4.** **(A)** Tile stitched fluorescence image of intranasally exposed WT mouse lung illustrating nonuniform microsphere distribution throughout entire left lung lobe. **(B)** Enlarged image of figure **(A)** to highlight nonuniform deposition at apex of lung. **(C)** Enlarged image of figure **(A)** to illustrate nonuniform deposition in middle area of lung. **(D)** Enlarged image of figure **(A)** to demonstrate nonuniform deposition in distal portion of lung. Arrows highlight microsphere clustering surrounding airways.

The results of this study illustrate the need for clinically representative models to study the health effects associated with aerosol exposure. As illustrated in Fig 3D, particle deposition due to administration via the exposure chamber demonstrated even distribution with microspheres being uniformly deposited throughout the parenchyma of the lungs. In contrast, mice which were exposed by using the traditional intranasal methodology, demonstrated nonuniform, sporadic deposition. This pattern of distribution is critical to consider when assessing experimental designs attempting to study exposure to aerosols and the possible associated health effects. As previously mentioned, our data revealed significant variation in intranasal administration. In preliminary preparation for this study, we used a method of one dose of 20 µl total, administering 5 µl per nostril at a time. In these mice, we noted varying results as some mice illustrated robust, nonuniform deposition with clustering around larger airways while others showed little to no microsphere deposition throughout the lung. Using this administration method, another cohort of mice showed very few microspheres making it through the respiratory tract and depositing in only the apex of the lungs, incapable of penetrating past the larger airways (Fig 6A and 6B). The clustering of microspheres around airways

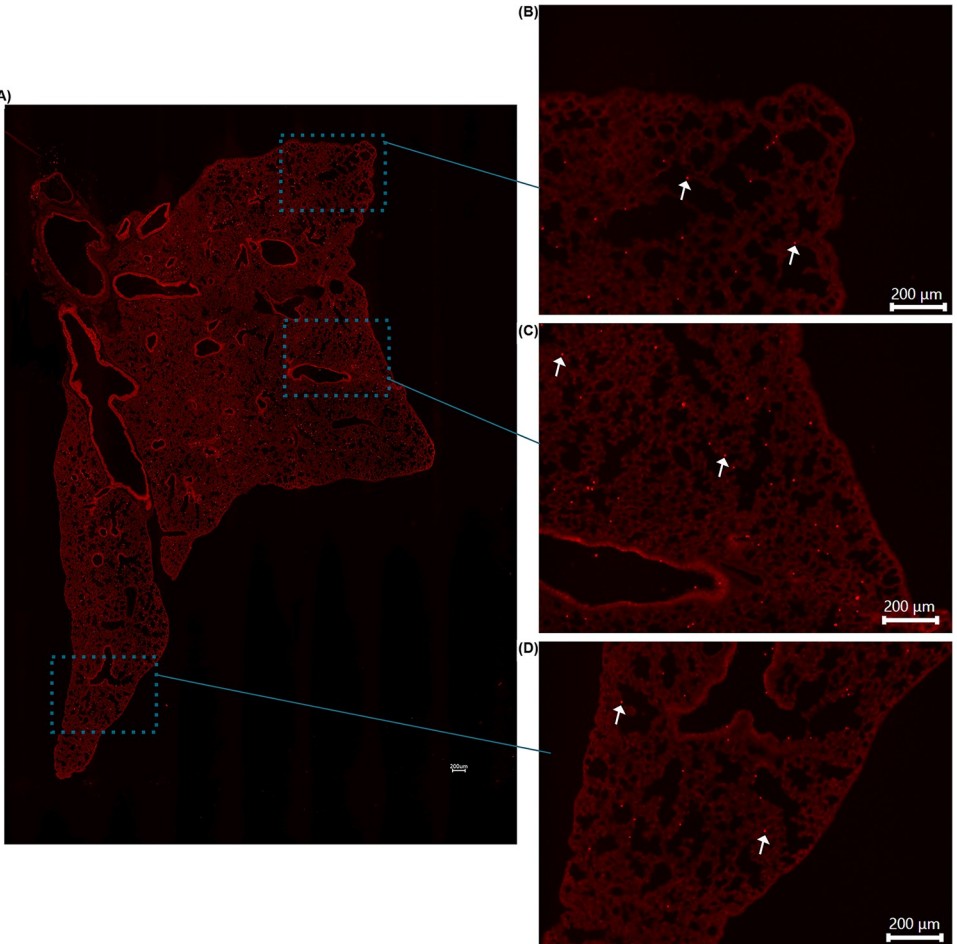

**Fig 5. (A)** Tile stitched fluorescence image of chamber exposed WT mouse lung illustrating uniform microsphere distribution throughout entire left lung lobe. **(B)** Enlarged image of figure **(A)** to highlight uniform deposition at apex of lung. **(C)** Enlarged image of figure **(A)** to illustrate uniform deposition in middle area of lung. **(D)** Enlarged image of figure **(A)** to demonstrate uniform deposition in distal portion of lung. Arrows highlight microspheres.

following intranasal administration suggests that this method of exposure could produce a response, which is not fully representative of the immune response in humans. This could ultimately mean that cellular responses will subsequently target the parenchyma and conducting airways of the lung differently leading to different areas of the lung itself being affected unevenly. The pattern of cellular recruitment is critical to understanding the behavior of the immune response. If cells are preferentially recruited to specific areas due to administration method, this may lead to varying immune responses when compared to an exposure done utilizing a method more similar to the natural route of inhalation such as those done utilizing an exposure chamber.

## Method of administration influences cellular recruitment pattern and behavior

To assess clinical implications of particle deposition, we exposed PGRP-S mice to 75 μg/mL of Lipopolysaccharide either intranasally or via the exposure chamber. During initial studies, it was difficult to estimate comparable concentrations for intranasal versus chamber aerosol

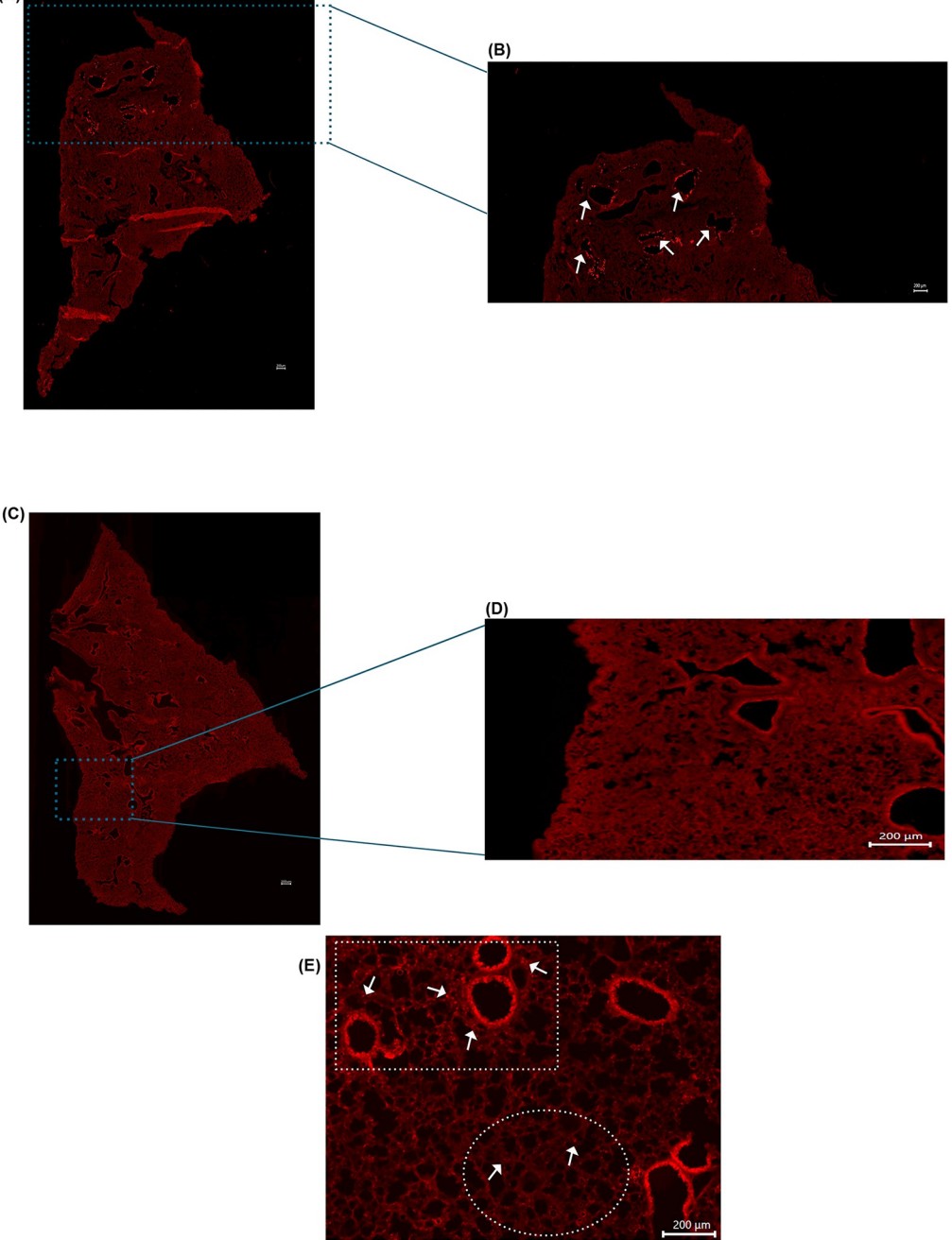

**Fig 6. (A)** Tile stitched fluorescence image of full left lobe of intranasally exposed WT mouse illustrating microsphere deposition at apex of the lung only. **(B)** Enlarged image of figure **(A)** to highlight deposition in apex of lung. Arrows highlight clustering of microspheres around airways. **(C)** Representative tile stitched fluorescence image of full left lobe of control WT mouse exposed to PBS intranasally **(D)** Enlarged image of figure **(C)**. **(E)** Fluorescence image of PGRP + mouse intranasally exposed to LPS illustrating preferential clustering around airways. White square highlights neutrophil clustering surrounding airways. White circle highlights minimal neutrophil deposition in parenchyma of lung. Arrows highlight PGRP+ cells recruited into the lung.

delivery due to the differences in how solutions entered the lungs; for example, while the input solution of 15 μg/mL of LPS was able to induce a response as an aerosol, when delivered intranasally this concentration was insufficient to induce cellular recruitment. Through

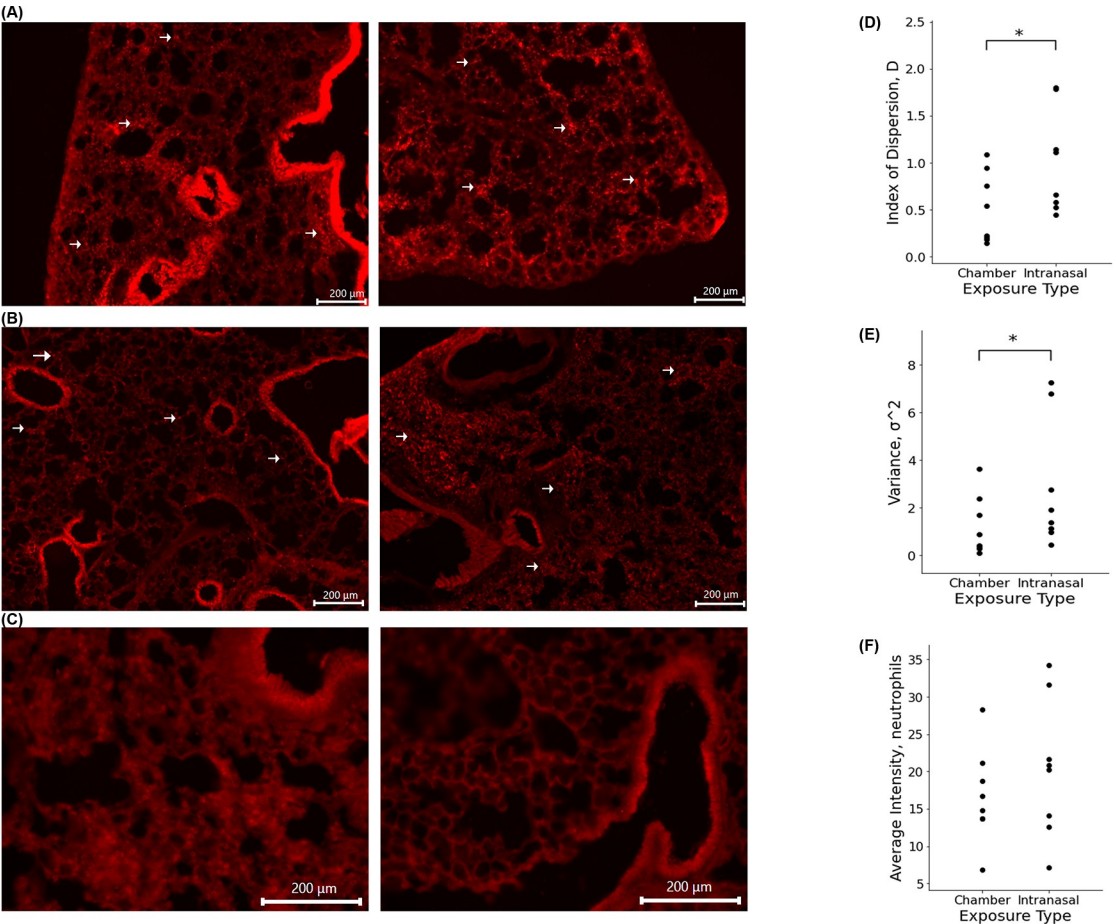

**Fig 7. (A)** Fluorescence images of cellular recruitment in PGRP+ mouse exposed to LPS via exposure chamber. **(B)** Fluorescence images of cellular recruitment in PGRP+ mouse exposed to LPS intranasally. Arrows highlight PGRP+ cells recruited into the lungs. **(C)** Fluorescence images of control lung. **(D)** Index of Dispersion of chamber and intranasally exposed PGRP+ mice. **p value = 0.009467 (E)** Raw variance of chamber and intranasally exposed PGRP+ mice. **p value = 0.042693 (F)** Average neutrophil density comparison of chamber and intranasally exposed mice. **p value = 0.251384**.

optimization, we found that 75 µg/mL was a concentration able to reliably induce significant cellular recruitment. We also found that administration must be done in one single bolus of 40 µl rather than delivering 5 µl/nostril at a single time; we previously found in the microsphere administration that small volumes often led to microspheres being embedded in the nasal passages and not entering the lungs. Upon administering LPS, we noted significant airway infiltration in lungs of mice both chamber exposed (Fig 7A) and intranasally exposed (Fig 7B). Quantitative analysis of dsRed fluorescence in the lungs of PGRP-S animals revealed results similar to those generated by microsphere administration (Fig 7D-7F). Analysis of neutrophil distribution throughout lung parenchyma of both exposure groups revealed that mice exposed to LPS intranasally demonstrated nonuniform distribution while chamber exposed mice exhibited uniform deposition. H&E histological staining and confocal imaging was done to verify establishment of inflammation and the cellular recruitment of neutrophils (Figs 8 and 9). Fig 8 illustrates a representative image of immune cell recruitment in intranasally exposed PGRP+ mice. To verify neutrophil recruitment, we conducted higher magnification confocal imaging on samples stained with a DAPI nuclear stain (Fig 9) to visualize multilobular nuclei of the recruited neutrophils. Mice exposed to LPS intranasally showed an average index of

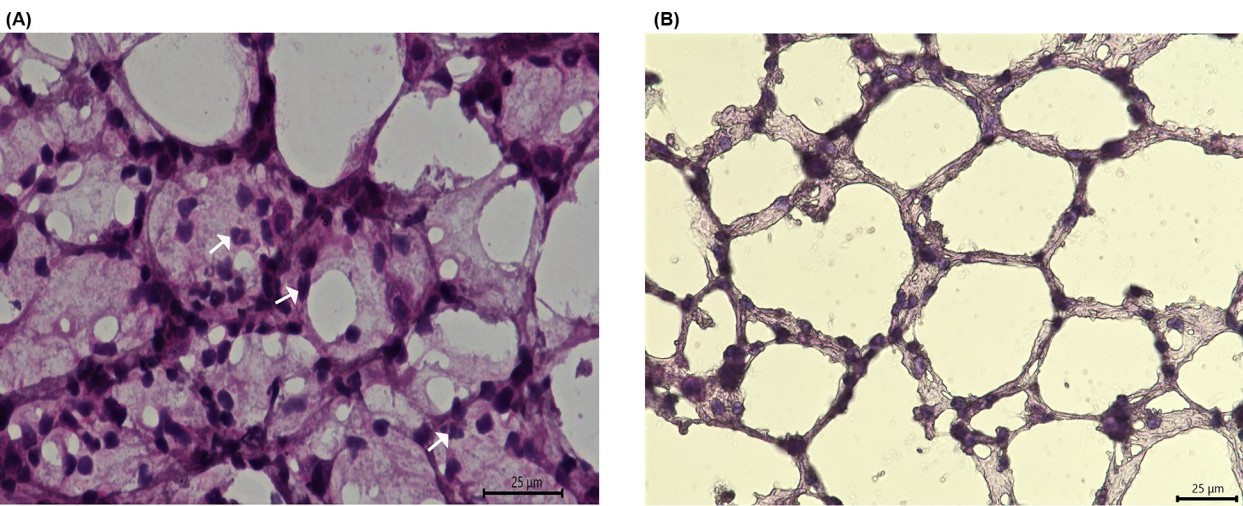

**Fig 8. Histological characterization of infiltrating cells. (A)** 60X hematoxylin and eosin stain of PGRP-S-dsRed model exposed to LPS intranasally. Arrows highlight infiltrating neutrophils. **(B)** 60X hematoxylin and eosin stain of WT control mice exposed to PBS intranasally.

dispersion of 0.981 while comparatively, chamber exposed mice illustrated an index of dispersion of 0.509. Along with a higher index of dispersion, intranasally exposed mice also demonstrated an increased variance when compared to those exposed via the chamber. Thus, our data suggest that the delivery of material into the lung influences the pattern of cellular recruitment.

In histological sections, intranasally exposed PGRP mice demonstrated dsRed+ neutrophil distribution similar to that of the corresponding microsphere animals. In these mice, we noticed sporadic recruitment of these cells, which showed clustering throughout the

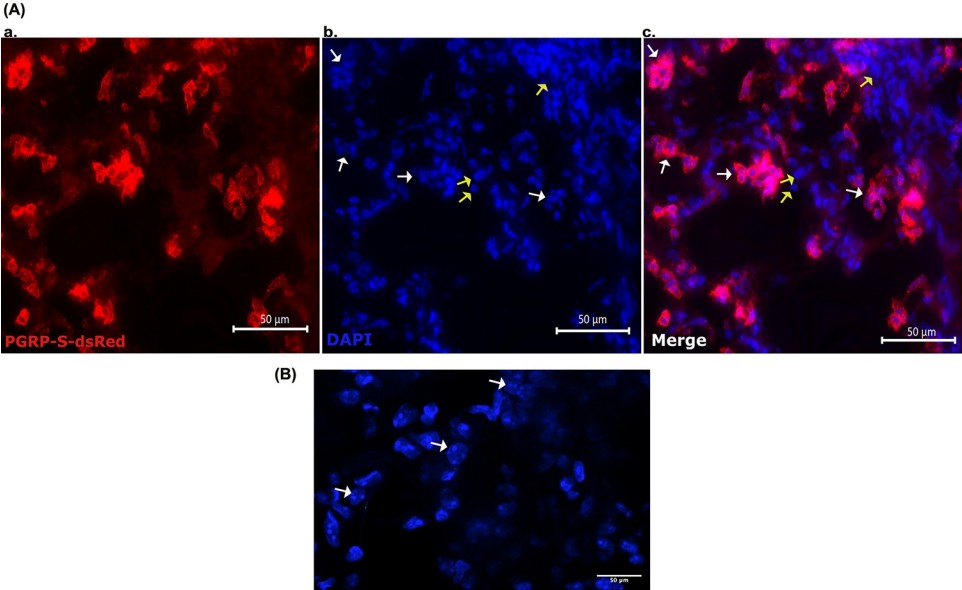

**Fig 9. Fluorescence Images of I.N. LPS exposed mice (Red = PGRP signal, Blue = DAPI. A) a.**60X PGRP **b.** 60X DAPI, yellow arrows signify epithelial cell nuclei; white arrows signify neutrophil nuclei **c.** 60X Merge of PGRP and DAPI. **B)** 63X magnified DAPI stain to show multilobular neutrophil nuclei.

parenchyma often leaving much of the tissue devoid of cellular recruitment. In contrast, chamber exposed mice demonstrated a distinct pattern in which cellular recruitment was uniform throughout the tissue, with no clustering and wide coverage throughout the parenchyma. This suggests that chamber exposed mice may have a broad inflammatory response due to the dispersed recruitment of neutrophils throughout the tissue while mice exposed intranasally elicit a varied immune response due to significant differences in tissue recruitment patterns between exposure routes.

The use of the PGRP-S-dsRed transgenic mouse model allowed for an in vivo analysis of our hypothesis demonstrating that the route of administration may affect biological responses to inhaled pathogens as well as other environmental toxins [12]. As highlighted in Fig 7D and 7E, this data provides clear support of the hypothesis and shows significant effects on cellular recruitment patterns. These results strongly suggest that the distribution of particles due to administration method significantly influences the behavior and recruitment pattern of immune cells. Interestingly, although the number of neutrophils recruited into the lungs was not significantly different between exposure groups (Fig 7F), as highlighted in Figs 6E and 7B, we noted clustering of cells around airways in intranasally exposed mice while the recruitment pattern of neutrophils in chamber exposed animals was uniform throughout the parenchyma. The clustering of neutrophils around medium and large airways was particularly striking; despite the fact that recruited neutrophils are intrinsically migratory within tissues, the intranasal administration still directed a clear pattern of preferential clustering. This data closely mimicked the pattern of particle deposition noted in the microsphere portion of this study, and illustrates the overall inconsistency of intranasal administration (Fig 6). The consequences of inflammatory neutrophil recruitment also will include distinct patterns of tissue damage and fibrosis induced by cellular inflammation. Accordingly, inconsistencies among methodologies should be considered before designing studies aimed to address immunological responses to inhaled materials, whether inert, immunostimulatory, or even infectious, to ensure that studies can provide clinically relevant physiological data.

## Clustering phenotype of I.N. exposed animals

We observed a distinct clustering pattern of beads and recruited neutrophils around the airways following intranasal instillation. In contrast, animals exposed to microspheres or LPS in the controlled chamber exhibited an even dispersion of cells and beads throughout the lung parenchyma. As highlighted in Fig 10, in the microsphere experimental group, 26.48% of particles deposited near an airway in the chamber-exposed animals whereas, 52.68% of the beads

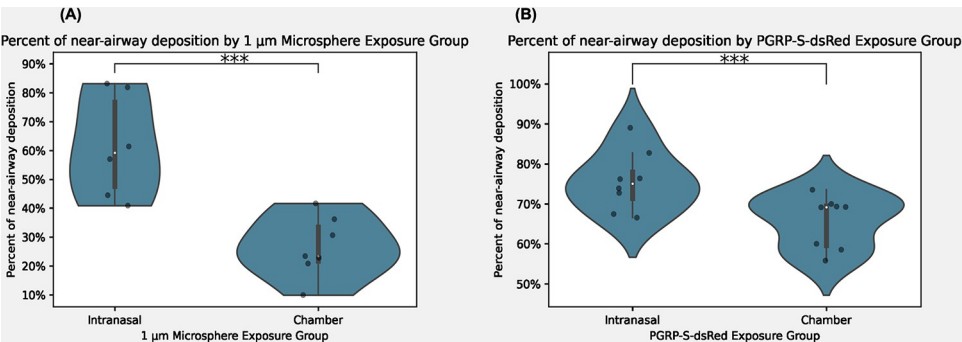

**Fig 10.** **(A)** Percent of near-airway deposition of microspheres in intranasal and chamber exposed animals. **p-value = 0.004496 (B)** Percent of near-airway deposition of recruited neutrophils in intranasal and chamber exposed animals. **p-value = 0.011466.**

exhibited a clustering phenotype within a 200 μm radius of a medium or large airway in the intranasally exposed animals. These values underscore the unique pattern of peri-airway clustering observed in the intranasally exposed animals. Similarly, PGRP-ds-Red transgenic mice exposed to LPS, 65.68% of cells clustered near the airways in the chamber-exposed animals, while 75.60% of cells demonstrated peri-airway deposition in the intranasal exposure group. Ultimately, this analysis revealed significant differences in distribution patterns between the two exposure methodologies, highlighting the remarkable clustering phenotype observed in the intranasally exposed animals.

## Discussion

This study provides evidence that the method of administering materials into the lung can significantly influence immunological responses, primarily due to variations in patterns of cellular recruitment. The lung is a complex organ that comprises two main distinct tissue compartments: the conducting airways and the respiratory airways or parenchyma, each characterized by unique cellular subsets. Consequently, disease processes involving inflammatory cell recruitment and associated cytokines are expected to have disparate effects within these compartments. As demonstrated in the current study, even in cases where inflammation triggers the recruitment of highly mobile cells like neutrophils, there are discernible variations in their recruitment patterns between these tissue compartments due to the administration method utilized. Therefore, it is imperative to carefully consider the divergent impacts of such recruitment on these distinct tissue compartments given the chosen method of delivery. This becomes particularly relevant in models investigating the effects of inhaled aerosols as well as disease model development, where models utilizing intranasal administration are prone to misrepresentation of the physiologically responses, especially those which depend on relevant impacts on the lung parenchyma.

In relation to the data presented here, other studies have aimed to highlight differences in immune response based on varying methods of administration. A recent clinical trial assessed the differences in immune response to cat allergen delivered via chamber exposure or nasal allergen challenge [16]. This study ultimately showed significant differences in overall magnitude of response with an increase in severity noted in the chamber exposed mice. In this study, they did not note significant differences in cytokine production; however, this measurement alone is not a true representation of overall immune response. A study by Hasegawa-Baba et al. sought to investigate the distribution of a test substance in rats utilizing variations of intratracheal administration [17]. This group found significant differences in distribution in regard to the technique used. In this study, they compared distribution of the substance under various conditions, altering the angle of the mouse, instillation speeds, as well as utilizing various devices. Intratracheal delivery techniques vary widely, so it may be difficult to compare results between studies utilizing this technique. The results of this study demonstrated that variation within technique of intratracheal administration alone produced significant differences in distributions of particles, highlighting the inconsistency of this method16. These results also raise the question of the reproducibility of models utilizing this method of administration. As in the present study, these inconsistencies are crucial as they suggest that techniques may introduce broad variation within cohorts of animals intended to address the same question. This is important to consider as it may affect reproducibility not only between exposures but also within a single exposure group. Although intratracheal administration bypasses the limitation of intranasal administration in which solution can get trapped in the nasal passages, there are also limitations to this method as it also administers a bolus of solution as opposed to aerosolization solution. Because of this, it does not simulate natural inhalation. In addition to

this, the delivery of a bolus of solution intratracheally may still result in nonuniform distribution and ultimately changes in cellular response patterns. To address this limitation for this method, further studies must be done to investigation distribution patterns and immune response.

## Conclusion

Current techniques such as I.N. administration, nose-only exposure chambers, and intratracheal administration have various limitations. In addition to lack of clinical relevance of these techniques, they also pose various technical limitations as some of these methods limit animal activity during time of exposure, making chronic exposures studies difficult to perform. This is a severe limitation as human response to aerosol exposure more often occurs by chronic rather than sporadic exposure to particulate matter. Our chamber design allows our laboratory to bypass previously mentioned limitations, allowing for generation of a clinically relevant exposure models in which we can investigate the health effects of various substances. This study not only highlights the inconsistencies of intranasal administration but more specifically provides compelling information about cellular behavior. The results of this study illustrate that intranasal administration of LPS, a robust immune stimulus, leads to localized cellular recruitment with characteristic peri-airway clustering. In contrast, exposure via aerosol chamber demonstrated uniform cellular recruitment throughout the lung tissue. These findings are important as they indicate biological consequences due to method of exposure, namely variations in cellular behavior such as more dispersed cell trafficking throughout the tissues to provide broad protection, pathogen sensing, and defensive responses, as well as consequent tissue damage and fibrosis.

The data produced by this study suggests that the use of intranasal administration methods in current aerosol studies which aim to investigate the relationship between aerosol exposure and associated health effects, may not accurately represent real-world human exposure. However, these results additionally indicate that intranasal delivery may serve as a suitable administration technique for studies focused on investigating models of airway toxicity or mucosal drug delivery. Ultimately, the findings of this study provide compelling evidence for the need for further studies to be done to ensure relevancy and accuracy of data attempting to address aerosol exposure and health implications employing the available methods of administration.

## Acknowledgments

The authors would like to acknowledge staff at UCR vivarium for their assistance with maintence and care of mouse colonies.

## Author Contributions

**Conceptualization:** Keziyah Yisrael, Qi Li, David R. Cocker, III, David D. Lo.

**Data curation:** Keziyah Yisrael, Ryan W. Drover, Malia L. Shapiro, Martha Anguiano, Nala Kachour, Qi Li, Emily Tran.

**Formal analysis:** Keziyah Yisrael, Ryan W. Drover, Qi Li, David D. Lo.

**Funding acquisition:** David D. Lo.

**Methodology:** Ryan W. Drover, Qi Li, David R. Cocker, III.

**Resources:** David D. Lo.

**Supervision:** David R. Cocker, III, David D. Lo.

**Validation:** Qi Li, David D. Lo.

**Visualization:** Malia L. Shapiro, David D. Lo.

**Writing – original draft:** Keziyah Yisrael.

**Writing – review & editing:** Ryan W. Drover, David D. Lo.

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
