## [Decision Letter · Decision Letter 0]

15 Jun 2023

PONE-D-23-15716

Route of Pulmonary Administration of Aerosols Significantly Affects Particle Deposition and Cellular RecruitmentPLOS ONE

Dear Dr. Lo,

Thank you for submitting your manuscript to PLOS ONE. After careful consideration, we feel that it has merit but does not fully meet PLOS ONE’s publication criteria as it currently stands. Therefore, we invite you to submit a revised version of the manuscript that addresses the points raised during the review process.

We look forward to receiving your revised manuscript.

Kind regards,

Abdelwahab Omri, Pharm B, Ph.D, Laurentian University

Academic Editor

PLOS ONE

Journal Requirements:

"The authors would like to acknowledge staff at UCR vivarium for their assistance with maintence and care of mouse colonies. 

The research presented in this publication was supported by the National Institute On Minority Health And Health Disparities of the National Institutes of Health under Award Number U54MD013368. The content is solely the responsibility of the authors and does not necessarily represent the official views of the National Institutes of Health."

"The research presented in this publication was supported by the National Institute On Minority Health And Health Disparities of the National Institutes of Health under Award Number U54MD013368 to DDL. The funders had no role in study design, data collection and analysis, decisions to publish, or preparation of the manuscript."

Reviewers' comments:

Reviewer's Responses to Questions

**Comments to the Author**

1. Is the manuscript technically sound, and do the data support the conclusions?

Reviewer #1: Yes

Reviewer #2: Partly

2. Has the statistical analysis been performed appropriately and rigorously? 

Reviewer #1: Yes

Reviewer #2: N/A

3. Have the authors made all data underlying the findings in their manuscript fully available?

Reviewer #1: Yes

Reviewer #2: Yes

4. Is the manuscript presented in an intelligible fashion and written in standard English?

Reviewer #1: Yes

Reviewer #2: Yes

5. Review Comments to the Author

Reviewer #1: This paper compares the distribution of flourescent microspheres administered to mice either in an exposure chamber or by intranasal instillation. The authors show that intranasal administration results in concentration of particles around the airways, whereas the chamber produces uniform distributrion of particles throughout the lung parenchyma. A very similar result was obtained using neutrophil reporter mice in an LPS model of either chamber exposure or intranasal administration.

This is a carefully done study and the authors have taken considerable care in their methods to ensure that their results are real. I don’t have any major criticisms, but do have two minor comments:

1. The Kogel study in which mice were exposed to cigarette smoke by whole body or nose only inhalation is cited as an example of the way in which administration by different routes leads to different results. Although this is true, what it really reflects in the Kogel study is pure nasal inhalation (ie, a good mimic of human smokers) vs licking of fir and ingestion (or direct skin uptake) of a toxin (ie, a poor model of human smokers). To me this is not a good model to cite for the authors’ hypothesis.

2. While the authors are quite correct that exposure chambers produce a much more realistic model than intranasal administration, there may be instances in which intranasal adminstration produces what you want: if what you are interested in is airway toxicity, then these data suggest that intranasal adminstration may be the route to go. It might be worth making this point in the discussion.

Reviewer #2: 1. The heading only depicting the pulmonary route whereas, intranasal route is also exposed in this study. The title need to modify as there is low connection between the title and manuscript content.

2. Author misunderstood the route of delivery vs delivery system. Aerosol is a delivery mechanism. Whereas, intranasal and pulmonary are the route of drug delivery. Rewrite the sentence no 28.

3. Are these ready made microspheres? or synthesized in lab? Mention the method of preparation of dye loaded microsphere.

4.Mention the Animal Ethical permission number. Is this approved protocol? What was the environmental conditions of the animals, body weight, grouping and sex?

5.I can see the schematic diagram of the chamber in the earlier publication of the authors. It will be good to insert the real image of chamber in the manuscript.

6.What is the delivery efficiency of the chamber? During exposure, how can author signifies that the particles will be delivered through pulmonary route? The delivery pattern is hard to understand.

7. Use standard unit in this manuscript

8.What is the Zeta potential of these microspheres? included it to understand the coagulation at airway. (line no 291)

9. It will be good if author summarize the result through a table form for better understanding of comparison. While reading, I found insistence in correlating one section with other.

6. PLOS authors have the option to publish the peer review history of their article (what does this mean?). If published, this will include your full peer review and any attached files.

Reviewer #1: No

Reviewer #2: **Yes: **Dr. Sujit Kumar Debnath

---

## [Author Response · Author response to Decision Letter 0]

12 Jul 2023

PONE-D-23-15716

Route of Pulmonary Administration of Aerosols Significantly Affects Particle Deposition and Cellular Recruitment

PLOS ONE

RESPONSE TO REVIEWERS

Reviewer #1

Thank you for your comments and suggestions. 

1. The Kogel study in which mice were exposed to cigarette smoke by whole body or nose only inhalation is cited as an example of the way in which administration by different routes leads to different results. Although this is true, what it really reflects in the Kogel study is pure nasal inhalation (ie, a good mimic of human smokers) vs licking of fir and ingestion (or direct skin uptake) of a toxin (ie, a poor model of human smokers). To me this is not a good model to cite for the authors’ hypothesis.

Response: Authors agree that this may not be the best reference and have removed this citation. 

2. While the authors are quite correct that exposure chambers produce a much more realistic model than intranasal administration, there may be instances in which intranasal adminstration produces what you want: if what you are interested in is airway toxicity, then these data suggest that intranasal adminstration may be the route to go. It might be worth making this point in the discussion.

Response: Lines 564-569 were included to address how intranasal administration may be a suitable technique for other interests and topics of study. 

Reviewer #2 

Thank you for your comments. 

1. The heading only depicting the pulmonary route whereas, intranasal route is also exposed in this study. The title need to modify as there is low connection between the title and manuscript content.

Response: Title was altered to “Route of Administration Significantly Affects Particle Deposition and Cellular Recruitment” address this concern. 

2. Author misunderstood the route of delivery vs delivery system. Aerosol is a delivery mechanism. Whereas, intranasal and pulmonary are the route of drug delivery. Rewrite the sentence no 28

Response: Line 28 was rewritten to be clearer. 

3. Are these ready made microspheres? or synthesized in lab? Mention the method of preparation of dye loaded microsphere.

Response: Expanded on section 2.1, the microspheres are ready to use and are from FluoSpheres, Molecular Probe. Dye is loaded utilizing a proprietary method from the vendor before receipt. 

4. Mention the Animal Ethical permission number. Is this approved protocol? What was the environmental conditions of the animals, body weight, grouping and sex?

Response: Yes, this protocol is approved. The animal ethical permission number is AUP #20210011 and is included in the methods section under mouse models.

5. I can see the schematic diagram of the chamber in the earlier publication of the authors. It will be good to insert the real image of chamber in the manuscript.

Response: Below is an image of the chamber system for reviewers; however, we do not believe that the set-up is appropriate to include in the manuscript as any pictures of the chambers will look too busy. 

6. What is the delivery efficiency of the chamber? During exposure, how can author signifies that the particles will be delivered through pulmonary route? The delivery pattern is hard to understand.

Response: The methods section was modified to include further discussion on the aerosol exposure chambers to address this question. The description includes further detail on the operation parameters, as well as a discussion on the various physical parameters involved in these mouse-aerosol interactions, as well as the merit of continuous and consistent (persistent) aerosol delivery providing a platform for evaluation of aerosol exposure that is the principal factor. Additionally, our laboratory has published multiple manuscripts referencing large datasets collected utilizing our aerosol exposure chamber system. Data provided via our chamber exposures are consistent and easily replicable, illustrating the consistency and efficacy of our exposure chambers. We have also been able to demonstrate varying biological responses due to different aerosol doses as well as different aerosol sources. 

a. Publications utilizing aerosol chamber:

i. “Aerosolized aqueous dust extracts collected near a drying lake trigger acute neutrophilic pulmonary inflammation reminiscent of microbial innate immune ligands”

https://www.sciencedirect.com/science/article/pii/S0048969722069820

ii. “Salton Sea aerosol exposure in mice induces a pulmonary response distinct from allergic inflammation” https://www.sciencedirect.com/science/article/pii/S0048969721035221

iii. “Establishment and characterization of a multi-purpose large animal exposure chamber for investigating health effects” https://pubs.aip.org/aip/rsi/article/90/3/035115/359553

iv. “Continuous inhalation exposure to fungal allergen particulates induces lung inflammation while reducing basal expression of innate immune molecules in the brainstem” https://www.ncbi.nlm.nih.gov/pmc/articles/PMC6053578/

7. Use standard unit in this manuscript

Response: Could the reviewer provide the line for which units are not in standard? If referring to the units for index of dispersion. This value is typically used with no units or arbitrary units since this value is a normalized ratio, as it is the reporting of the ratio between two statistical measures (described here by the Statistical Engineering Division of the National Institute of Standards and Technology; https://www.itl.nist.gov/div898/software/dataplot/refman2/auxillar/ind_disp.htm). When applied in other applications, it is similarly reported unitless and described as a normalized ratio of measures as opposed to a physical measure; we refer to one of these manuscripts to describe index of dispersion by Rasmus Nielsen entitled “Roobustness of the Estimator of the Index of Dispersion for DNA Sequences”. This paper should help to provide background. 

8. What is the Zeta potential of these microspheres? included it to understand the coagulation at airway. (line no 291)

Response: The zeta potential of the microspheres has not been measured by the manufacturer. Our laboratory does not have the proper equipment to measure the zeta potential or electrostatic charge of the particles in aerosol suspension; however, aerosol suspensions are monitored during exposures utilizing a scanning mobility particle sizer (SMPS, including Series 3080 Electrostatic Classifier and Ultrafine Condensation Particle Counter 3776, TSI) and a laser aerosol spectrometer to assist in maintaining stable and repeatable environmental exposure conditions. With this monitoring of the aerosol suspension, our instruments reported an equal size particle distribution with aggregation of the microspheres in the aerosol suspension. 

9. It will be good if author summarize the result through a table form for better understanding of comparison. While reading, I found insistence in correlating one section with other.

Response: Summary table added at the end of the results section, beginning line 478.

---

## [Editor Report · Decision Letter 1]

18 Jul 2023

Route of Administration Significantly Affects Particle Deposition and Cellular Recruitment

PONE-D-23-15716R1

Dear Dr. Lo,

We’re pleased to inform you that your manuscript has been judged scientifically suitable for publication and will be formally accepted for publication once it meets all outstanding technical requirements.

Kind regards,

Abdelwahab Omri, Pharm B, Ph.D, Laurentian University

Academic Editor

PLOS ONE

---

## [Editor Report · Acceptance letter]

21 Jul 2023

PONE-D-23-15716R1 

Route of Administration Significantly Affects Particle Deposition and Cellular Recruitment 

Dear Dr. Lo:

I'm pleased to inform you that your manuscript has been deemed suitable for publication in PLOS ONE. Congratulations! Your manuscript is now with our production department. 

Kind regards, 

on behalf of

Dr. Abdelwahab Omri 

Academic Editor

PLOS ONE